# DNA Karyometry for Automated Detection of Cancer Cells

**DOI:** 10.3390/cancers14174210

**Published:** 2022-08-30

**Authors:** Alfred Böcking, David Friedrich, Martin Schramm, Branko Palcic, Gregor Erbeznik

**Affiliations:** 1Institute of Cytopathology, University Clinics, 40225 Düsseldorf, Germany; 2AstraZeneca, 80636 München, Germany; 3Department of Cytopathology, Institute of Pathology, Heinrich-Heine University, 40225 Düsseldorf, Germany; 4Cancer Imaging Department, BC Cancer Agency, Vancouver, BC V7H2X4, Canada; 5Noki Medical d.o.o., 1000 Ljubljana, Slovenia

**Keywords:** automated microscope-based screening, oral smears, Fanconi anemia, supervised machine learning, computer assisted diagnosis, grading prostate cancer, cancer cell detection

## Abstract

**Simple Summary:**

Cancers have to be microscopically established before they can be treated adequately. This can be performed on cell and/or tissue samples. Smears or sedimentations of cells on glass slides are used to microscopically screen large number of specimens for the presence of cancer cells. Such screenings require highly specialized personnel, which is not available in most countries. We present a microscope-based scanner which is able to identify cancer cells in smears from the oral cavity or in body cavity fluids. In addition, using the same system, the degree of malignancy of prostate cancer can be determined. A combination of image features of stained nuclei and their DNA-content is used to raise a suspicion of malignancy. Nuclear diagnostic classifiers were trained by an expert using supervised machine learning. All device-proposed diagnoses can be verified by a specialist. The overall percentage of correct device-derived diagnoses on oral smears was 91.3% as compared to 75.0% for conventional, subjective investigation.

**Abstract:**

Background: Microscopical screening of cytological samples for the presence of cancer cells at high throughput with sufficient diagnostic accuracy requires highly specialized personnel which is not available in most countries. Methods: Using commercially available automated microscope-based screeners (MotiCyte and EasyScan), software was developed which is able to classify Feulgen-stained nuclei into eight diagnostically relevant types, using supervised machine learning. the nuclei belonging to normal cells were used for internal calibration of the nuclear DNA content while nuclei belonging to those suspicious of being malignant were specifically identified. The percentage of morphologically abnormal nuclei was used to identify samples suspected of malignancy, and the proof of DNA-aneuploidy was used to definitely determine the state malignancy. A blinded study was performed using oral smears from 92 patients with Fanconi anemia, revealing oral leukoplakias or erythroplakias. In an earlier study, we compared diagnostic accuracies on 121 serous effusion specimens. In addition, using a blinded study employing 80 patients with prostate cancer who were under active surveillance, we aimed to identify those whose cancers would not advance within 4 years. Results: Applying a threshold of the presence of >4% of morphologically abnormal nuclei from oral squamous cells and DNA single-cell or stemline aneuploidy to identify samples suspected of malignancy, an overall diagnostic accuracy of 91.3% was found as compared with 75.0% accuracy determined by conventional subjective cytological assessment using the same slides. Accuracy of automated screening effusions was 84.3% as compared to 95.9% of conventional cytology. No prostate cancer patients under active surveillance, revealing DNA-grade 1, showed progress of their disease within 4.1 years. Conclusions: An automated microscope-based screener was developed which is able to identify malignant cells in different types of human specimens with a diagnostic accuracy comparable with subjective cytological assessment. Early prostate cancers which do not progress despite applying any therapy could be identified using this automated approach.

## 1. Introduction

All cancers must be microscopically diagnosed and classified before they can be treated. This can be performed subjectively on cells or tissues under a light microscope cytologically or histologically by skilled personnel, mostly pathologists. Well-educated cytotechnicians may assist in prescreening cytological specimens. While the bioptic acquisition of tissue needs scalpels or biopsy needles and local anesthesia, that of cells needs neither. They can be obtained noninvasively either from body fluids (cerebrospinal fluid, serous effusions, sputa, bronchial secretions, or urine) or by brushing mucosal surfaces (conjunctival, oral, pharyngeal, laryngeal, bronchial, bile ducts, cervical, or perianal). Using fine-needle aspiration, biopsy cells can also be obtained from inner organs (thyroid, salivary glands, lungs, lymph nodes, liver, pancreas, kidneys, or prostate). The compliance of patients for a cytological clarification of a given suspicious lesion (e.g., an oral leukoplakia) is, therefore, higher than that for histological ones. Because of their noninvasiveness and painlessness, cytological tests can also be used for mass screening. The most well-known and acknowledged cytologic screening is that for cervical cancer according to Papanicolaou. Screening of oral leukoplakias in patients with increased oral cancer risk, such as in Fanconi anemia, of sputum in patients with increased risk of lung cancer, and of urine in patients with increased risk of bladder cancer is well established. Applying adjuvant methods, such as immunocytochemistry, fluorescence in situ hybridization (FISH), or DNA cytometry, can help to increase diagnostic and typing accuracy [1]. In addition to the fact that most cancer cell-positive cytological diagnoses are subsequently validated by histological ones on tissues of the resected respective lesions, the diagnostic accuracy reaches high levels [2,3,4,5]. Furthermore, objective and prognostically valid grading of the malignancy of special types of cancer, such as the prostate, measuring the DNA content of thousands of cancer cells in enzymatic cell separation specimens, is another task that needs automation of diagnostic cytometry [6].

Cytopathology represents a subdiscipline of pathology and requires special diagnostic training. Screening cytological specimens needs high levels of concentration, in addition to being time-consuming and tiring. Specially trained cytotechnicians assist especially in screening repetitive smears, but the final cancer-cell-positive diagnoses must be validated by pathologists. Diagnostic accuracy greatly depends on the education and concentration of cytopathologists and cytotechnicians. It is, thus, very variable. At least in Germany, cytological investigations are less well paid by insurance companies and, accordingly, patients. This has a negative effect on their popularity among pathologists. Furthermore, the availability of skilled cytotechnicians is sinking, among others because most schools for their education have been closed in this country. In most countries of the third world, they are not available at all. Thus, the benefits of cytological screening for cancer cells and clarification of lesions suspicious for cancer do not reach most patients worldwide.

Thus, the availability of a computerized microscope which will be able to automatically screen cytological specimens from various sites of the human body for the presence of cancer cells and their precursors is very desirable. It should not replace cytotechnicians or cytopathologists but save their time and focus their sophisticated work on solving delicious diagnostic questions and confirming positive or suspicious diagnoses, e.g., proposed by the machine.

For about 25 years, automated Papanicolaou test screening devices have been commercially available. Two automated systems are in widespread use: the ThinPrep imaging system from Hologic (Marlborough, MA, USA) and the FocalPoint GS imaging system from Becton and Dickinson (Franklin Lakes, NJ, USA). They present a limited number of microscope fields of view with the highest likelihood of the presence of cellular abnormalities for further evaluation to cytotechnologists. These devices do not intend to detect single morphologically abnormal cells or to definitely identify cancer cells. They simply allow faster subjective screening [7].

The Cyto-Savant has been developed in joint cooperation between the Oncometrics Imaging Corporation and the Cancer Imaging Department of the British Columbia Cancer Agency in Vancouver, Canada [8]. It was finally distributed by Motic, China, as MotiSavant. Similar to our solution, it is based on nuclear DNA measurements. However, it does not comprise an expert-trained, artificial intelligence (AI)-based morphologic classification of abnormal nuclei based on supervised machine learning. It has been used for screening smears for cervical, oral, and lung cancer. Screening for cervical cancer reached a sensitivity of 89.58% and a specificity of 56.25% for the detection of HSIL and higher lesions [9,10].

Earlier developments were the Leyden Texture Analysis System (LEYTAS) developed by the University of Leiden, the Netherlands and provided by Leitz, Germany [11], the PapNet, provided by Neuromedical Systems [12], and the Cyto-Biologic Electronic Screening System, CYBEST, developed by the Toshiba Research and Development Center, Tokyo, Japan [13].

The optimal staining used for automated microscope detection of cancer cells is a controversial issue. As DNA aneuploidy is a 100% specific marker for malignancy [14], we prefer DNA-specific pararosaniline or thionine in Feulgen staining. In addition to the fact that it allows nuclear DNA measurements with a CV of <3% in TV image cytometry [15], it allows recognizing diagnostically relevant details of nuclear chromatin pattern that cannot be recognized with other stainings (Figure 1).

There is no doubt that malignancy arises in the nuclei of cells, not within their cytoplasm and manifests itself in specific nuclear shapes and chromatin patterns that can be used diagnostically. It is, therefore, reasonable that malignancy can mostly be detected in nuclei from cells earlier than in tissues [16]. Staining of cytoplasm is, thus, mostly not needed for the detection of malignant cells.

Feulgen-stained nuclei can be subjectively and objectively separated into those derived from normal epithelial/mesothelial cells, abnormal epithelial/mesothelial cells, lymphocytes, granulocytes, fibroblasts, and macrophages. Artefacts in cytological specimens can be subdivided into nuclear doublets and overlaps, defocused, lytic nuclei, and non-nuclear objects (dirt). The crucial diagnostic task of a computerized microscope for the detection of cancer cells is to identify nuclei (a) suitable for internal calibration of DNA content in c-values (e.g., from normal epithelial or mesothelial cells and fibroblasts), (b) that are suspicious of a malignant transformation (cancer cells) and their precursors (dysplasias or borderline lesions), both called abnormal nuclei, and (c) from inflammatory cells and different types of artefacts for removal.

While the term “diagnostic DNA cytometry” means the diagnostic assessment of nuclei based on DNA measurements only, the term “diagnostic DNA karyometry” means the combination of an automated diagnostic morphometric classification of nuclei and DNA measurements on morphologically abnormal ones [4].

### 1.1. Prostate Cancer Grading

Active surveillance of locally confined, low-grade prostate cancers represents a therapeutic strategy, based on monitoring only, avoiding side-effects and complications of invasive strategies. Yet, patients have to be sure that their individual cancer will not progress despite waiver of therapy. Conventional subjective histologic grading according to Gleason does not provide sufficient safety [17]. Objective and valid grading of the malignancy of prostate cancers has repeatedly been shown to add relevant prognostic information [6]. Instead, objective measurement of nuclear DNA in a representative number of cancer cells yields a prognostically valid DNA grade of malignancy [18,19] (Figure 2). Four different DNA grades of malignancy can be differentiated (Figure 2). Due to the known heterogeneity of prostate cancers, all biopsies containing cancer cells must be processed, and a high number of nuclei must be measured each time. This is time-consuming for manual DNA cytometry. A microscope-based device for automated DNA measurements of thousands of cancer cell nuclei is, therefore, required.

### 1.2. Screening Effusions

The average sensitivity of conventional cytological evaluation of serous effusions in order to identify cancer cells is only about 58% [4]. In many countries of the world, insufficient specially trained cytotechnicians and cytopathologists to microscopically evaluate slides from serous effusions are available. Furthermore, subjective screening of cytological slides is time-consuming. Therefore, a semiautomated microscopical procedure to diagnostically scan respective slides is desirable.

### 1.3. Oral Smears

Brush biopsies of suspicious oral leukoplakias and erythoplakias represent a noninvasive alternative to invasive scalpel biopsies for the assessment of their dignity. With a sensitivity of 97.7% and a specificity of 84.5%, they are suitable for oral cancer screening in high-risk groups, such as smokers, betelnut chewers, or Fanconi anemia patients [5]. Due to the fact that their diagnostic interpretation needs highly qualified personnel, such as cytopathologists and cytotechnicians, which are rare worldwide, an automated diagnostic assessment of smears from oral brush biopsies is desirable.

Thus, the aim of this article is to present so far obtained results applying the computerized slide-scanner EasyScan-AI for automated diagnostic screening for cancer cells in serous effusions and oral smears and for grading the malignancy of prostate cancers, using the technology of diagnostic/prognostic DNA karyometry.

## 2. Materials and Methods

### 2.1. Preparation of Specimens

Specimens suitable for automated microscopical screening for the presence of cancer cells should represent monolayers. Conventional smears from brushings or fine-needle aspirations are not adequate, because of too much cellular overlap or irregular distribution of cells. Direct cytocentrifugations on glass slides can be used from body fluids, such as effusions or urine. Brushings from mucous membranes (oral, laryngeal, bronchial, bile ducts, cervical, or perianal) are subjected to alcoholic fixation combined with mucolysis and subsequent cytocentrifugation on glass slides. Even pre-stained slides can be used, as the stains will be removed during Feulgen staining. The procedure for enzymatic cell separation of histologic specimens was described elsewhere [18]. Feulgen staining according to Feulgen and Rossenbeck has also been described several times elsewhere [20]. Feulgen staining kits are commercially available. To maintain a specific temperature of 25.0 °C is crucial. Machines, capable of automated Feulgen staining, equipped with a temperature-controlled, HCl-resistant cuvette are commercially available (e.g., Resostain 23-F, Resolab, Bad Qeynhausen, Germany). The microscope-based scanning devices, described below, can handle thionine and pararosaniline as DNA-specific stains.

### 2.2. Scanning Devices

#### 2.2.1. MotiCyte-Auto

The microscopic hardware of the MotiCyte (Motic China Group, Ltd., Xiamen, PR China) software solution [18,19] consists of a motorized Motic BA 610 microscope with a 40× objective (NA of 0.65) and a MotiCam CCD color camera 285 A with 1360 × 1024 pixel resolution. A scanning stage for single slides was connected with an autofocus stepper device. Effective pixel density sampling within the slide was between 0.1 μm and 0.25 μm in the *x*- and *y*-directions. The system’s light response was linear within the color range of the stain used.

The software used was that of MotiCyte screener (Version no. 2.3, creator David Friedrich, Aachen, Germany) with digital nuclear classifiers [21,22,23,24,25], specified for effusions [4], prostate cancers [6], and oral smears. The classifiers were trained on Feulgen/pararosaniline-stained slides to automatically discriminate normal and abnormal nuclei, lymphocytes, granulocytes, and artefacts (e.g., lytic and defocused nuclei) in oral smears. In effusions, normal and abnormal mesothelial nuclei, those from lymphocytes, granulocytes, and macrophages, and lytic nuclei, defocused nuclei, and doublets were discriminated. In prostate cancers, nuclear preparations from digested tissue, cancerous cells, lymphocytes, granulocytes, fibroblasts, and artefacts were differentiated. Scanning a whole slide took between 40 and 60 min.

Details of applied nuclear segmentation and correction of different optical errors (diffraction, glare, and shading) were described in detail by Würflinger et al. [23].

Scene-specific focusing is a major concern in the automated analysis of cytological specimens, as, unlike histological sections, the objects of interest often lay in different levels. On the one hand, the technique of cytocentrifugation facilitates this problem. On the other hand, either one optimal focus plane has to be found per slide location by stepping through at least seven optical planes (realized in MotiCyte). In EasyScan-AI, for the identification of the most precise image of each individual object per microscope image, the mostly focused image out of seven optical levels is chosen by software (Figure 3).

The morphologic digital nuclear classifiers for DNA karyometry of effusions were trained and tested on a gold standard of an annotated database of images from 54,374 Feulgen-stained nuclei derived from nine different slides. The KNN classifier reached a correct classification rate of 88.11%. A total of 35,920 nuclei from nine patients were used for training the oral cervical classifier, reaching a correct classification rate of 89.02%. A total of 47,982 nuclei from nine patients were used for training nuclear classification in prostate cancer specimens, reaching an accuracy of 98.81% [21]. The primary subjective classification of nuclei was performed by an experienced cytopathologist (A.B.). While 18 different morphometric features were used, 11 of them described the nuclear morphology of nuclei, four were pixel values of the brightfield images, and three represented textural information. The random forest classifier was applied [21]. For other tumors, this classifier performed comparably to support vector machines and outperformed k-nearest neighbor classifiers, conventional decision trees, neural networks, or the AdaBoost classifier [21]. Furthermore, 95.2% of abnormal nuclei were correctly identified by the effusion-classifier, and 2.0% of artefacts were misclassified as abnormal. Additionally, 3.5% of abnormal nuclei were erroneously classified as artefacts, and 4.2% of objects that automatically classified as artefacts were abnormal nuclei [21].

Internal calibration of nuclear DNA values was performed with nuclei from normal mesothelial cells in effusions, from normal epithelial cells in oral smears and from fibroblasts in prostate cancer specimens. Outliers below and above the respective mean value were eliminated automatically, until CV values <4% were reached.

Before algorithmic diagnostic interpretation of DNA distribution of abnormal nuclei, the adequacy of their morphologic classification was checked on the respective image gallery and reclassified if necessary. This could take 3 min per slide.

The applied algorithms for the detection of DNA stemline and single-cell aneuploidy as specific marker for malignant cells were consented by the European Society for Analytical Cellular Pathology (ESACP) [25,26,27,28]. Stemlines <1.8c, >2.2c and <3.6c, >4.4c were interpreted as DNA stemline aneuploidy, and the occurrence of nuclei >9c was interpreted as single-cell aneuploidy. In cases in which DNA aneuploidy as a 100% specific marker of malignancy could be detected in the population of morphologically abnormal nuclei, the presence of cancer cells could definitively be assumed. We have not yet observed false-positive diagnoses applying this algorithm in different types of specimens [22]. In cases without DNA aneuploidy but an increase in the percentage of morphologically abnormal nuclei (e.g., >4% or >0.75% in effusions), a suspicion of malignancy can be raised [4].

#### 2.2.2. EasyScan-AI

EasyScan (Figure 3) represents a professional microscope-based digital scanner that was developed to scan histopathologic and cytopathologic slides for the purpose of digital storing, transferring, browsing, and finally evaluating histological and cytological specimens diagnostically. It can be remotely controlled (telepathology). It is equipped with a high-precision motion control system, a high-resolution, 5M color CMOS camera, and high-resolution digital imaging. The 20× lens with NA of 0.75 allows a resolution of 0.52 μm/pixel and the 40× lens with 0.75 NA of 0.26 μm/pixel. The device can be equipped with automated slide loaders for six or 100 slides.

The time required to scan and diagnostically evaluate one Feulgen-stained cytocentrifugation specimen at 40× primary magnification currently is about 20–30 min.

A special software allows selecting from seven images per stack and slide location the mostly focused ones per object and constructing a new image out of these. This results in an extension of depth of focus according to the depth of autofocus. As a result, the yield of brightly represented nuclei per field of view is much higher; thus, more representative results can be obtained.

As we were not satisfied with the above mentioned rates of correct classifications of different types of nuclei and other objects by MotiCyte-software (version 2.3, creator David Friedrich, Aachen, Germany [18], we added further training sets from additional microscope slides (nine each from oral smears, effusions, and prostate cancers). Our aim was to reach an accuracy of classification for each type of object >95%, applying supervised machine learning. The results of each rescan using a new classifier were checked subjectively on the digital image galleries (A.B.). Misclassified objects were specifically reclassified. Thus, eight rounds of retraining the MotiCyte digital nuclear classifiers were performed. For the final three training rounds, the training sets were enriched by reclassified objects. Thus, self-learning, a relevant aspect of artificial intelligence of nuclear classifiers, was performed. The diagnostic and prognostic interpretation remained the same as described for MotiCyte-auto.

A microscope-based scanner with slide loader with a capacity of up to 10 specimens, a high-resolution color CMOS camera, a multistep autofocus, and AI-based diagnostic software were used. Specimens were stained with pararosaniline, specifically for DNA. In a first step, objects are automatically classified into eight different diagnostic classes (normal and abnormal epithelial or mesothelial nuclei, lymphocytes, granulocytes, lytic nuclei, defocused nuclei, doublets, and artefacts). Internal calibration of DNA content in c-values was performed with morphologically normal epithelial/mesothelial or fibroblast nuclei. In a second step, nuclear DNA was measured with high precision (CV < 4%) in the population of morphologically abnormal nuclei. A specific marker of malignancy was the presence of single-cell aneuploidy (9c exceeding events) or DNA stemline aneuploidy according to standards of the European Society for Analytical Cellular Pathology (ESACP) [24,25]. In the absence of DNA aneuploidy, an increased percentage (>5 or >4%) of morphologically abnormal nuclei were used to raise a suspicion for the presence of cancer cells. The software for diagnostic nuclear classification is based on random forest classifiers and supervised machine learning performed in up to eight rounds of retraining by an experienced cytopathologist (A.B.). An accuracy of >95% for correct classification of different classes of objects in terms of nuclei was achieved. Operators may check the correctness of nuclear classifications on image galleries and that of diagnostic interpretations on DNA histograms.

### 2.3. Opportunistic Oral Cancer Screening (New Results)

A cohort of liquid-based brush biopsy-based oral preparations was taken from that described by Velleuer et al. [5]. The study was conducted in accordance with the Declaration of Helsinki, and approved by the Western Institutional Review Board of the USA (study number 1139633) and by the ethics committee of the medical faculty of the Heinrich Heine University in Düsseldorf, Germany (study number 4168). Written informed consent was obtained from all participants. Accordingly, our current study consisted of consecutive 92 cytospin preparations from 40 patients between 2016 and 2018 with available follow-up data. The process of visual inspection of the oral cavity, photo-documentation of the suspect lesions (leukoplakias, erythroplakias, mixed lesions, ulcers/erosions), the brushing process, the preparation of liquid-based cytologic samples, and the staining with pararosaniline according to Feulgen were described by Velleuer et al. [5]. A skilled pediatrician (E.V.) performed all oral inspections and took brush biopsies of macroscopically suspicious lesions. Visible ones (leukoplakias or erythroplakias) were sampled and processed as described by Velleuer et al. [5]. Subsequently, DNA-specific staining with pararosaniline according to Feulgen was performed using a staining machine, as described by Velleuer et al., [5]. The EasyScan-AI as described above was used, applying a 40× objective. The total area containing oral cells was scanned per slide. Automated classification of objects was performed into nuclei from normal and abnormal epithelial cells, lymphocytes, granulocytes, nuclear doublets, defocused nuclei, and artefacts. Scanning of the slides was performed by a trained engineer (G.E.) blinded to the follow-up data. After scanning, a cytopathologist (A.B.) checked the results of automated nuclear classifications (especially into nuclei from normal with respect to abnormal epithelial cells) on respective image galleries (Figure 4 and Figure 5). A total of 63 nuclei were, thus, processed per screen. Nuclear doublets, defocused nuclei, and artefacts were removed.

Indicators of malignancy were the occurrence of ≥1 9c exceeding events (single-cell aneuploidy) and/or DNA-stemlines > or <3.6–4.4c (stemline aneuploidy) [24,25]. The occurrence of >5% in contrast to 4% of abnormal epithelial nuclei was interpreted as suspicious for malignancy. In the final assessment, the occurrence of DNA aneuploidy and/or the occurrence of >5% in contrast to ex post 4% of abnormal nuclei was used as indication of a cancer-cell-positive specimen and, thus, assessed as a “positive result”.

The scanning results were compared to the same follow-up reference standard as described in Velleuer et al. (2020) [5]. A negative (benign) or low-grade OED histological diagnosis within 6 months or a negative clinical course within 2 years of oral examination and cytology in the same region was defined as negative reference standard.

A histological diagnosis of an SCC or high-grade OED (including moderate and severe OED) within 6 months of oral examination and cytology in the same region was defined as positive reference standard. A positive cytological diagnosis or the detection of DNA aneuploidy with consistent clinical course (i.e., SCC therapy, definite imaging, or palliative care) was also defined as a positive reference standard (Table 1a).

In four slides, DNA aneuploidy was detected but the respective histology was negative; cancer was proven in the immediate neighborhood in a palliative situation. These diagnoses were rated as correct positive in a separate evaluation. In another case with proven DNA aneuploidy but negative cytology, a resection of cancer was performed shortly before smear taking. This diagnosis was also rated as correct positive (Table 1b).

Indicators of malignancy were the occurrence of ≥1 9 c exceeding events (single-cell aneuploidy) and/or DNA stemline aneuploidy [28]. The occurrence of >5% in contrast to 4% ex post of abnormal epithelial nuclei was interpreted as suspicious for malignancy. In the final assessment, the occurrence of DNA aneuploidy and/or >5% in contrast to ex post >4% of abnormal nuclei was used as an indication of a cancer cell-positive slide and, thus, as “positive” result.

Cytological diagnoses were established at the Department of Cytopathology, Institute of Pathology, University Hospital Düsseldorf, Germany (head: Dr. Matin Schramm). Diagnostic categories were negative, suspicious, urgently suspicious, and positive for cancer cells, as described by Velleuer et al. (2020) [5]. The cytological diagnoses were performed and already published in the large context of the latter study. However, the cytological results of the 92 liquid-based preparations described here were extracted, and the diagnostic accuracies were recalculated, taking all but negative cytology as positive test results.

### 2.4. Grading Prostate Cancer, Reviewed from [6]

Enzymatic cell separation specimens from needle-biopsies of 80 patients with clinical stage T1a/b and T2a/b prostate cancers, Gleason scores 6 and 7, as described by Böcking et al. (2017) [6], were used. Pararosaniline/Feulgen-stained sedimentation slides were automatically scanned using a MotiCyte-auto (Motic, Xiamen, China). The development of digital nuclear classifiers to specifically identify nuclei from normal fibroblasts for internal reference of the DNA amount, from cancer cells, granulocytes, lymphocytes, and artefacts, was described by [29]. A mean number of 4669 cancer cell nuclei were measured per slide. Internal calibration was performed with nuclei from 268 fibroblasts in the mean. DNA histograms were separated into peridiploid (DNA-grade 1), peritetraploid (DNA-grade 2), x-ploid (DNA-grade 3), and multiploidy (DNA-grade 4) (Figure 2) [18]. Follow-up information was provided by the respective treating urologists.

### 2.5. Screening Effusions, Reviewed from [4]

A total of 121 consecutive cytocentrifugation slides from patients presenting with serous effusions were investigated using a MotiCyte-auto system (Motic, Xiamen, China), as described by Böcking et al. (2018) [4]. After Feulgen re-staining of sedimentation smears with pararosaniline and internal calibration with 3734 normal mesothelial nuclei in the mean, the DNA content of 3026 automatically detected abnormal nuclei in the mean was used for diagnostic analysis. The occurrence of nuclei >9c (9c exceeding events) and/or the presence of aneuploid DNA stemlines < or >1.8–2.2c or 3.6c–4.4c was used as an indicator for the presence of malignant cells and assessed as test-positive [26,27]. Follow-up was taken from the respective patients’ files.

## 3. Results

### 3.1. Performance Standards

The European Society for Analytical Cellular Pathology (ESACP) has published performance standards for diagnostic DNA image cytometry imprints to be tested on rat liver imprints [28]. These demand a maximum CV of lymphocytes of 5% and a maximum CV of diploid hepatocytes of 5%. Berger-Fröhlig (2018) [15] tested the fulfilment of these standards by the MotiCyte device. For that purpose, >50 lymphocytes as internal reference nuclei and >50 diploid (2c) hepatocyte nuclei in 4 × 4 rat liver imprints of two rats were measured. The mean CVs of lymphocyte IODs were 3.05% and 2.64%, while those of 2c hepatocytes were 4.36% and 4.42% for rat 1 and rat 2, respectively.

### 3.2. Oral Cancer Screening, New Results

Mean numbers of normal and abnormal nuclei per slide, as well as uncorrected and corrected results, including their respective standard deviations, are presented in Table 2. A mean number of 3426 normal epithelial nuclei were detected in smears from all patients, of which 2970 remained after subjective removal of the artefacts (Figure 4). A TOTAL OF 148 abnormal epithelial nuclei were detected in all patients, from which 120 remained after elimination of the mentioned artefacts. A total of 357 abnormal nuclei were automatically classified in all smears from cancer-patients, of which 313 remained after subjective corrections (Figure 5). In smears from patients without cancer, 117 abnormal nuclei were automatically classified, of which 44 remained after subjective correction.

Table 1a,b presents the dependence of four variables with respect to diagnostic accuracy of the algorithm, used for diagnostic interpretation and their combination: (1) occurrence of 9c exceeding events single-cell aneuploidy), (2) occurrence of abnormal DNA stemlines in c (stemline aneuploidy), (3) single-cell and DNA stemline aneuploidy together, (4) occurrence of >5% abnormal epithelial nuclei as a percentage of all normal nuclei, (5) occurrence of >4% abnormal epithelial nuclei as a percentage of all normal nuclei, (6) combination of DNA single-cell and stemline aneuploidy with occurrence of >5% of abnormal nuclei, and (7) combination of DNA single-cell and stemline aneuploidy with the occurrence of >4% abnormal nuclei. Table 1a uses the originally communicated follow-up diagnoses as a reference standard, while Table 1b uses a modified one. In four slides, DNA aneuploidy was detected but the respective follow-up within the original definitions was negative, whereby cancer was proven in the immediate neighborhood in a palliative situation. These diagnoses were rated as correct positive in a separate evaluation. In another case with proven DNA aneuploidy but negative cytology and histology of low-grade OED, a resection of cancer was performed in the clinical history at the site of oral brushing. This diagnosis was also rated as correct positive (Table 1b). In 11 of 25 slides with positive follow-up (defined by the modified reference standard), DNA aneuploidy alone was found. These were considered as “cancer cell-positve”. In 15 out of 17 slides from patients with cancer, >5% in contrast to 4% of abnormal nuclei were detected. All but one DNA aneuploid slides revealed >5% in contrast to >4% morphologically abnormal nuclei. This specimen showed stemline and single-cell aneuploidy, but the proportion of abnormal nuclei was 3.55%. This was because of the high number of 7043 detected corrected normal epithelial cells that “diluted” the proportion of abnormal nuclei. In four slides with positive follow-up, >4% and >5% abnormal nuclei were detected but no DNA aneuploidy. These were considered as “suspicious for cancer cells” and, thus, as test-positive.

Our original hypothesis supposed a threshold of >5% as morphometric evidence of malignant cells, resulting in a sensitivity of 64% at a specificity of 100%, PPV of 100%, and NPV of 88.2%. Lowering this threshold to <4% as morphometric evidence of malignant cells resulted in a sensitivity of 72% at a specificity of 98.5%, PPV of 94.7%, and NPV of 90.4% (Table 1b).

The overall diagnostic accuracy (ODA) of automated DNA karyometry applying our originally assumed diagnostic algorithm, comprising a threshold of >5% of abnormal nuclei and the original follow-up diagnoses, was found to be 84.8%, in contrast to 85.9% for >4% and 76.1% for manual cytology. Applying the modified follow-up diagnoses, the ODAs were 90.2% for >5% abnormal nuclei, 91.3% for >4% abnormal nuclei, and 75.0% for manual cytology.

### 3.3. Prostate Cancer Grading, Reviewed from [6]

Progression of cancers occurred in 37.5% if upgrade of any inclusion criterion and in 18.8% if only PSA-DT <36 months or upstaging to pT3 was used as evidence. DNA grade 1 was diagnosed in 40% of cases. The sensitivity, specificity, and negative predictive value of Gleason score, assessed by the reference pathologist and DNA karyometry, were 20.0%, 86.7%, and 76.5%, in contrast to 85.0%, 51.0%, and 90.6% if upgrade of any criterion of inclusion was used as evidence of progression, and 23.5%, 87.3%, and 80.9% and 100%, 50.8% and 100% if only PSA-DT <36 months and/or upstaging to pT3 were used, respectively. No progression of prostate cancers within 4.1 years was observed in patients with DNA grade 1 (Figure 2).

### 3.4. Screening Serous Effusions, Reviewed from [4]

Using DNA aneuploidy as a marker for malignancy in 121 smears, 59 with cancer cells resulted in a sensitivity of 76.4% as compared to 88.5% for subjective cytology, both at a specificity of 100%. The relationship between automatically classified morphologically abnormal and all mesothelial nuclei was found to identify cancer cell–positive slides at 100% sensitivity but 70% specificity. The time effort for an expert to control the results was reduced to the verification of a few nuclei with exceeding DNA content on a computer monitor image gallery, which can be accomplished within about 3 min.

## 4. Discussion

### 4.1. Microscope-Based Scanners

The purpose of microscope-based scanners in the field of diagnostic cytology is to assist in screening repetitive cytological slides for the presence of cancer cells. They can identify and present regions of interest on a cytological slide that have to be assessed under a microscope in detail by skilled personnel, or they can definitively identify individual malignant or dysplastic cells (i.e., their respective nuclei). The latter solution will be more time-saving and will require less skilled personnel. We herein present such a solution.

If nuclear DNA content, i.e., DNA aneuploidy, is used as a specific marker of malignant cells as diagnostic parameter, as in our solution, a DNA-specific stain, such as that proposed by Robert Feulgen [21], is required. Thionine or pararosaniline can be applied. This requires a temperature (25 °C)-controlled acid (HCl)-resistant cuvette. The procedure can be performed automatically using acid-resistant staining machines comprising a temperature-controlled cuvette. Re-staining of samples according to Papanicolaou is possible for a subsequent subjective inspection of problematical slides.

The microscope-based scanner should comprise a device for automated loading between 10 and 100 slides. The microscope should be equipped with a 40× high-NA objective. Scanning should be fast, completed within a few minutes per slide. A mechanical stepper should allow catching several images per field of view at different levels of focus as cells in cytological slides are mostly found at different *z*-levels. The color CMOS camera should allow high spatial and photometric resolutions (Figure 1).

Diagnostic DNA karyometry represents the combination of automated morphometric classification of nuclei into those of different cell types, including reference cells, normal and suspicious nuclei, and DNA image cytometry. While the percentage of morphologically abnormal nuclei within a slide (e.g., >4%) can be used to raise suspicion of malignancy, the presence of DNA aneuploidy (single-cell and/or stemline aneuploidy) detected in the fraction of morphologically abnormal nuclei can be used as a specific marker of malignant cells. The combination of both features provides a means for a higher diagnostic accuracy. In our series of 92 oral smears, DNA aneuploidy alone recognized only 40% of malignant clones; adding nuclear morphometry with the occurrence of >4% abnormal nuclei raised the sensitivity to 60%. Another task of diagnostic nuclear morphometric classification, in addition to the identification of reference cell nuclei, is to remove diagnostically irrelevant ones (e.g., lymphocytes, granulocytes, nuclear doublets, defocused nuclei, or artefacts) from diagnostic DNA cytometry. The universal algorithms suitable for diagnostic interpretation of nuclear DNA histograms have been internationally consented and published by the European Society for Analytical Cellular Pathology, ESACP [26,27,28,30]. The percentage of abnormal nuclei suitable for raising a suspicion of malignancy depends on the type of tissue, the fixation and staining of specimens, and the nuclear classifier applied. For effusion specimens scanned with a MotiCyte-auto, we found a suitable threshold of >0.75% [4]; for oral smears, we herein reported >4% to be useful. Cytotechnicians or cytopathologists can easily control the precision of nuclear classifications using image galleries and of diagnostic interpretations on DNA histograms. A cytopathological diagnosis should only be issued after a skilled operator has reviewed all nuclei on image galleries classified as morphologically abnormal and their respective DNA histogram in order to confirm DNA euploidy or aneuploidy.

### 4.2. Screening Oral Smears for Cancer Cells

Automated classification of normal and abnormal epithelial nuclei from oral smears (Figure 1, Figure 4 and Figure 5) amounted to 86.7% in contrast to 80% correctness after subjective control on image galleries. Using automatically classified and subjectively controlled in contrast to corrected nuclei of squamous epithelial cells as the internal reference to define the normal 2c value, controlled morphologically abnormal nuclei for analysis, and DNA aneuploidy as a marker of malignancy, a sensitivity of 44% was found for the detection of malignant nuclei according to our modified follow-up. When adding the percentage of morphologically abnormal nuclei of 5% in contrast to 4% as an indicator of nuclei suspicious for malignancy, sensitivity raised to 64% in contrast to 72%. Respective specificities were 100% and 98.5% (Table 1b).

DNA stemline and single-cell aneuploidy served as a highly (close to 100%) specific marker of malignancy. An increased percentage of morphologically abnormal nuclei served as a marker for suspected malignancy. Thus, the increased percentage of morphologically abnormal nuclei per slide helps to increase the diagnostic sensitivity of diagnostic DNA karyometry of oral smears. The reason for false-negative diagnoses most likely was not a paucity of cells but sampling errors, as eight of nine false-negative slides contained sufficient (>1500) normal epithelial nuclei (up to 8090). False positives, based on DNA aneuploidy as a marker, did not occur. The only false suspicious slide contained 7.8% morphologically abnormal nuclei.

The sensitivity and specificity of cytological assessment of oral smears according to an updated Cochrane systematic review are 90% and 94%, respectively [5]. The sensitivity of subjective cytomorphological investigation of oral smears from 737 Fanconi-anemia patients was reported by Velleuer et al. (2020) [5] to be 97.7%, with a specificity of 84.5%. The combination with manual DNA cytometry yielded a sensitivity of 100% and a specificity of 92.2%. Thus, one advantage of our current method is its high specificity.

The estimated mean time needed for visual control and correction of nuclei automatedly classified as abnormal on an EasyScan-AI screen is 3 min, while a subjective microscopical screen of a liquid-based cytological specimen requires about 6 min. For that purpose, well-educated and experienced personnel are required (cytotechnicians) that are not available anywhere worldwide.

### 4.3. Grading the Malignancy of Prostate Cancer

It is well known that a significant percentage of screening-detected prostate cancers do not progress, even without therapy, thus being suitable for the conservative strategy of active surveillance (45.3% in [31]). They may be identified by their low histological grade of malignancy on core biopsies. Unfortunately, the subjective assessment of histological grades of malignancy of prostate cancers according to Gleason is insufficient (45.7% according to [32] Relying on this parameter in order to choose active surveillance instead of active therapy is, therefore, critical. In a previous study on 80 patients, we could prove the superior prognostic validity of an automatically obtained DNA grade of malignancy (Figure 2) in early-stage prostate cancer patients [5]. During the follow-up period of 4.1 years, the probability to exclude a progression of an untreated, localized prostate cancer under active surveillance was 100% for objective DNA karyometry, but only 80.9% for the subjective microscopic Gleason score [6]. This means that patients with Gleason score 6 and 7 prostate cancers, who reveal an objectively assessed DNA grade 1 of malignancy, can safely rely on this conservative strategy. The time-consuming selection of fibroblast nuclei for internal reference of the normal 2 c DNA content and of thousands of cancer cell nuclei for analysis can now be automatically performed by a microscope-based scanner such as the EasyScan-AI, specially trained to identify these nuclear types and to derive internationally consented DNA grades of malignancy [18,20]. Realizing the known heterogeneity of prostate cancers, the fact that thousands of nuclei can be analyzed using such a device (we found 4669 in the mean) additionally resulted in a better representativity of measurements as compared with manual ones.

Low numbers of cytological samples so far investigated with the new technology of automated diagnostic/prognostic DNA karyometry limit the representativity of our respective data. Further studies considering more representative patients in order to better substantiate diagnostic accuracies as compared to subjective diagnoses/prognoses are required.

### 4.4. Screening Effusions for Cancer Cells

Serous effusions are very frequent events in patients from internal medicine. Approximately 40% of these malignant cells can be found, very often unexpectedly [32]. In these cases, they are the first evidence of a malignant tumor. Thus, all serous effusions have to be microscopically investigated for the presence of cancer cells. Yet, not only is this time-consuming, but there are not enough skilled personnel worldwide to perform this task with sufficient diagnostic accuracy. The average sensitivity of subjective cytological screening of serous effusions without adjuvant methods is 58% [3]. Therefore, a device that is able to automatically scan cell sedimentation slides from effusions for the presence of cancer cells is very welcome. We recently published that the precursor of our EasyScanAI, specifically trained to classify nuclei from different cell types for subsequent DNA measurements in effusions, is able to perform such screening [4]. In this study we investigated slides from 136 patients with known follow-up. While manual DNA cytometry yielded only 34 normal mesothelial cells in the mean as an internal reference per slide, automated DNA karyometry found 3734 and 68 more abnormal nuclei per slide. Our reported sensitivity of 76.4% at a specificity of 100%, therefore, allows the application of our method in daily routine.

### 4.5. Remote Applications

As online access to the microscopical scanning device is possible, reviewing the obtained microscopical and DNA cytometric results can be performed remotely by skilled experts in cytopathology, independent from the place of screening.

### 4.6. Future Applications

Following the training of nuclear classifiers for further applications on suitable sets of cytological specimens, e.g., cerebrospinal fluid, bronchial secretions, sputa, or urine, further applications will be possible, providing respective tests of diagnostic accuracy.

## 5. Conclusions

We reported herein a microscopical device able to automatically scan cytological specimens from oral smears and serous effusions for the presence of cancer cells and their precursors (dysplasias), and to grade the malignancy of prostate cancer cells. The respective hardware and software used for this diagnostic/prognostic DNA karyometry was described. The used diagnostic algorithm is based on the combination of the detected percentage of morphologically atypical nuclei (>4%) and of DNA aneuploidy. The former parameter is used to raise a suspicion of malignancy, while the latter is used for its definitive diagnosis. Diagnostic nuclear classifiers were trained in several rounds by an experienced cytopathologist, using the random forest classifier strategy. The operator is asked to check the adequacy of automatically classified atypical nuclei and of the resulting DNA histogram before a diagnosis is proposed. In a blinded study on oral smears from 92 patients with Fanconi anemia, the device reached an overall diagnostic accuracy of 84.8% as compared to 76.1% when using manual cytology. Ex post lowering the threshold of abnormal nuclei to >4% increased the overall diagnostic accuracy to 91.3%.

## Figures and Tables

**Figure 1 cancers-14-04210-f001:**
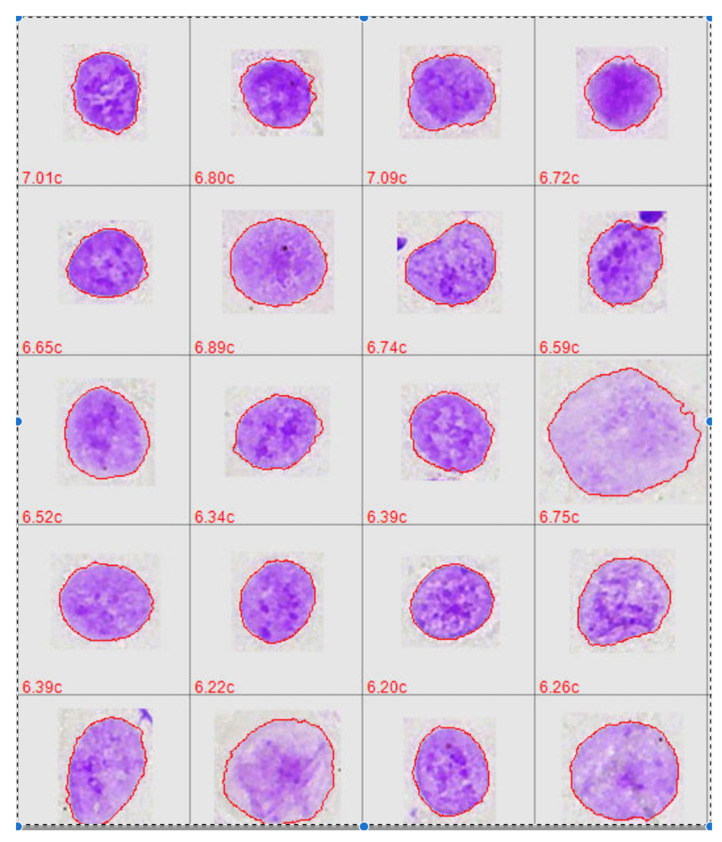
Nuclear images of automatically classified 20 Feulgen/pararosaniline-stained oral squamous carcinoma cells. Chromatin patterns and nuclear shaped can well be recognized.

**Figure 2 cancers-14-04210-f002:**
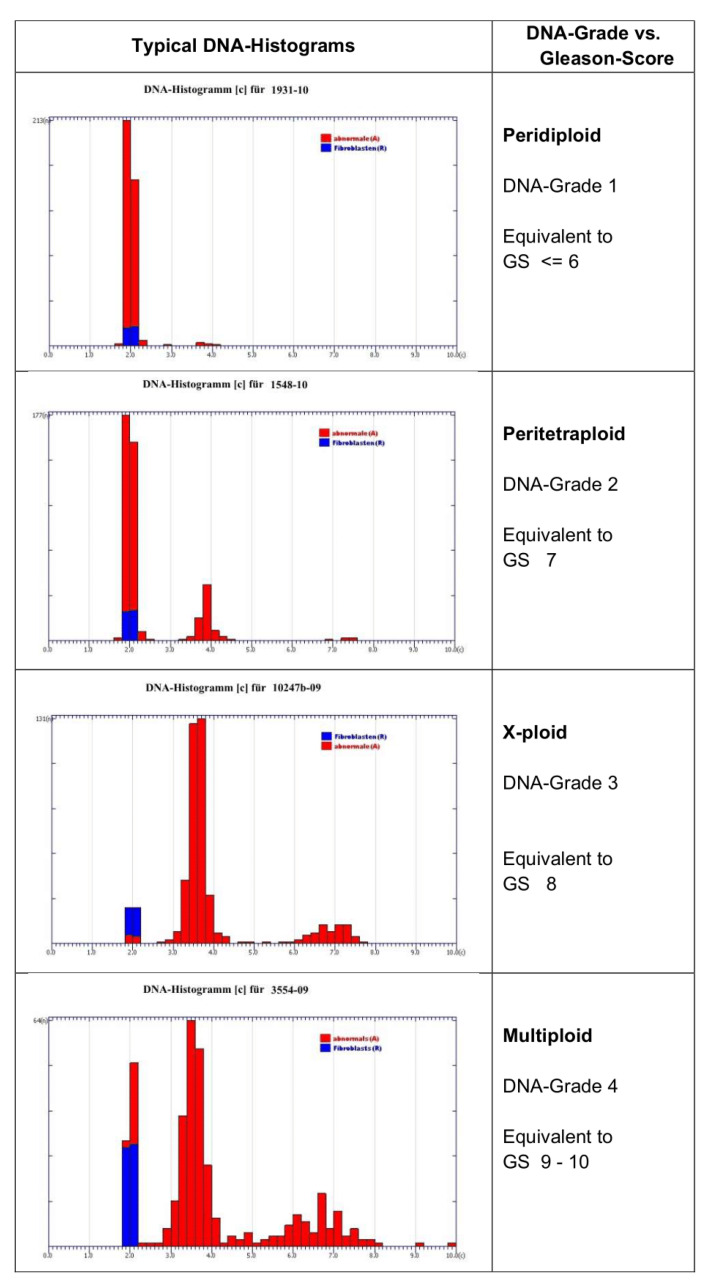
DNA histograms of four prognostically different DNA grades of malignancy in prostate cancers (grade 1 = peridiploid, grade 2 = peritetraploid, grade 3 = x-ploid, grade 4 = multiploidy), from Böcking et al. [18].

**Figure 3 cancers-14-04210-f003:**
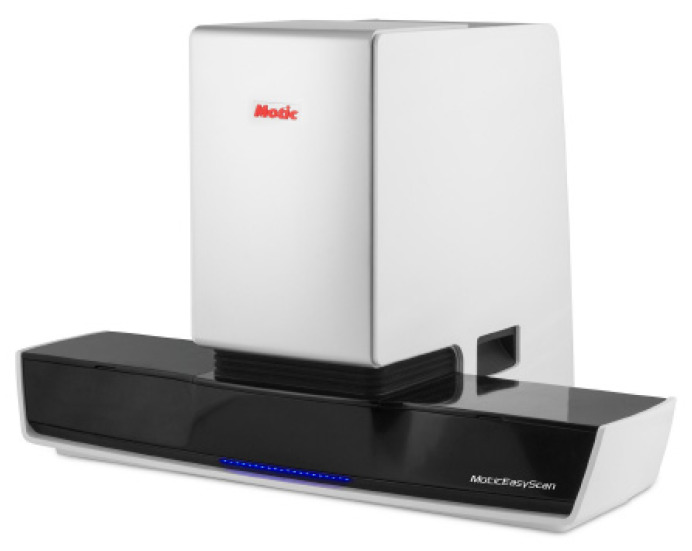
Microscopical scanner EasyScan-AI from Motic for automated scanning of 10 slides.

**Figure 4 cancers-14-04210-f004:**
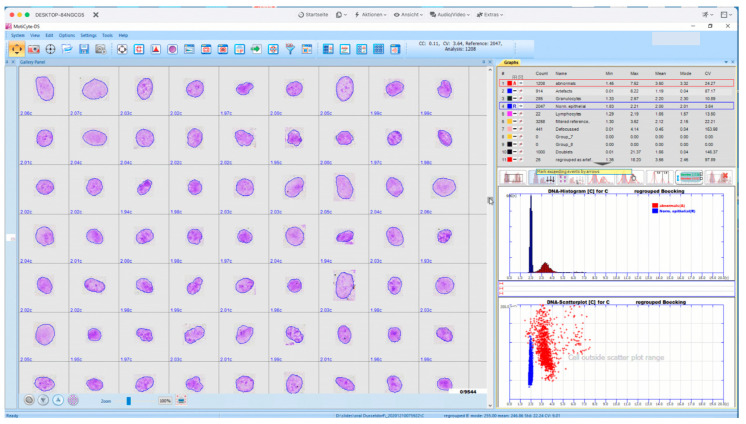
Interactive screen from an EasyScan-AI, presenting an image gallery of 63 automatically classified normal epithelial, Feulgen-stained nuclei from an oral smear, including their respective DNA contents (**left**), the respective DNA histogram and scattergram (area vs. DNA) (**right below**), and numbers plus different statistical features of detected five nuclear types (**right above**).

**Figure 5 cancers-14-04210-f005:**
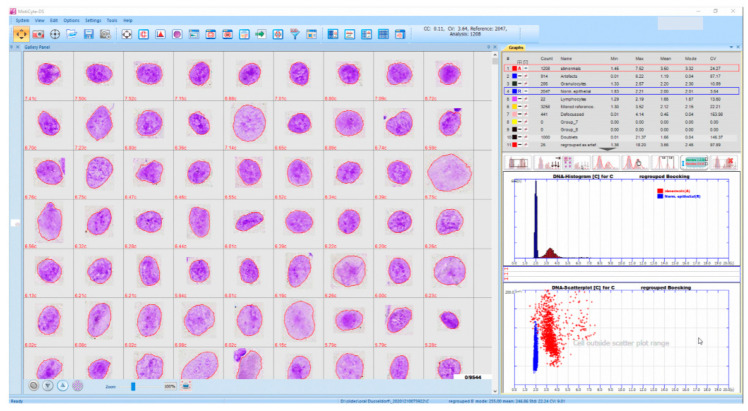
Interactive screen from an EasyScan-AI, presenting an image gallery of 63 Feulgen-stained, automatically classified cancer cell nuclei and their respective DNA contents (**left**), the respective DNA histogram and scattergram (area vs. DNA) (**right below**), and numbers plus statistical features of detected five nuclear types (**right above**).

**Table 1 cancers-14-04210-t001:** (**a**) Accuracy of different diagnostic algorithms in 92 oral smears from patients with FA, scanned with EasyScan-AI. Original follow-up. (**b**) Accuracy of different diagnostic algorithms in 92 oral smears from patients with FA, scanned with EasyScan-AI. Modified follow-up. **Bold**: original hypothesis.

(a)								
	DNA Single-Cell Aneuploidy 9cEE	DNA STL Aneuploidy	DNA Single-Cell and STL Aneuploidy	Abnormals >5%	Abnormals >4%	DNA-Aneuploidy and >5% Abnormals	DNA- Aneuploidy and >4% Abnormals	Cytology
Sensitivity	4/20 = 20%	6/20 = 30%	7/20 = 35%	10/20 = 50%	12/20 = 60%	11/20 = 55%	13/20 = 65%	16/20 = 80%
Specificity	69/72 = 95.8%	70/72 = 97.2%	68/72 = 94.4%	67/72 = 93.1%	66/72 = 91.7%	67/72 = 93.1%	66/72 = 91.7%	54/72 = 75%
PPV	4/7 = 57.1%	6/8 = 75%	7/11 = 63.6%	10/15 = 66.7%	12/18 = 66.7%	11/16 = 68.8%	13/19 = 68.4%	16/34 = 47.1%
NPV	69/85 = 81.2%	70/84 = 83.3%	68/81 = 84%	67/77 = 87%	66/74 = 89.2%	67/76 = 88.2%	66/73 = 90.4%	54/58 = 93.1%
Overall diagnostic accuracy						78/92 = 84.8%	79/92 = 85.9%	70/92 = 76.1%
(**b**)								
	**DNA Single-Cell Aneuploidy** **9cEE**	**DNA STL Aneuploidy**	**DNA Single-Cell and STL-Aneuploidy**	**Abnormals >5%**	**Abnormals >4%**	**DNA-Aneuploidy and >5% Abnormals**	**DNA- Aneuploidy and >4% Abnormals**	**Cytology**
Sensitivity	7/25 = 28%	8/25 = 32%	11/25 = 44%	15/25 = 60%	17/25 = 68%	16/25 = 64%	18/25 = 72%	18/25 = 72%
Specificity	67/67 = 100%	67/67 = 100%	67/67 = 100%	67/67 = 100%	66/67 = 98.5%	67/67 = 100%	66/67 = 98.5%	51/67 = 76.1%
PPV	7/7 = 100%	8/8 = 100%	11/11 = 100%	15/15 = 100%	17/18 = 94.4%	16/16 = 100%	18/19 = 94.7%	18/34 = 52.9%
NPV	67/85 = 78.8%	67/84 = 79.8%	67/81 = 82.7%	67/77 = 87%	66/74 = 89.2%	67/76 = 88.2%	66/73 = 90.4%	51/58 = 87.9%
Overall diagnostic accuracy						**83/92 = 90.2%**	84/92 = 91.3%	69/92 = 75.0%

**Table 2 cancers-14-04210-t002:** Mean values (x) and standard deviations (sd) of classified nuclei in 92 EasyScan-AI-scanned oral smears from patients with Fanconi anemia, 25 with positive follow-up (SCC or high-grade OED). SCC: squamous cell carcinoma, OED: oral epithelial dysplasia; pos FU: positive follow-up; neg FU: negative follow-up; STL: stemline; c: content.

Type of Object	x	sd
Normal epithelials uncorrected in all smears	3426.00	3396.24
Normal epithelials corrected in all smears	2970.43	3186.11
Abnormal epithelials uncorrected in all smears	148.41	255.05
Abnormal epithelials corrected in all smears	119.65	223.80
Abnormal epithelials in all smears with pos FU uncorrected	356.96	378.58
Abnormal epithelials in all smears with pos FU corrected	313.64	328.89
Abnormal epithelials in all smears with neg FU uncorrected	118.68	93.30
Abnormal epithelials in all smears with neg FU corrected	43.70	93.30
% abnormal epithelials in all smears	5.25	13.30
9cEE in all smears with pos FU	6	-
x STL in c in allsmears with pos FU	3.47	-
>5% abnormal epithelials in all smears with pos FU	68	-
>4% abnormal epithelials in all smears with pos FU	76	-
% aneuploid STLs in all smears with pos FU	56	-

## Data Availability

More detailed data concerning the object numbers found in scans of 92 oral smears including machine and follow-up diagnoses can be provided by Alfred Böcking (alfred.boecking@web.de).

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
