# Peer review of "DNA Karyometry for Automated Detection of Cancer Cells"

_cancers, 2022, doi:10.3390/cancers14174210_

Round 1

Reviewer 1 Report

- Excellent paper, congratulation

- too much information: please focus the paper on screening only, thus shorten and streamline the manuscript accordingly (e.g. remove chapter dealing with grading of prostate samples); avoid redundencies in M&M chapter described previously in published studies 

Author Response

Reviewer 1:

  • We now have deleted redundant methodological details from Velleuer et al. (2020), as proposed.
  • We would like to maintain the short chapter on “DNA-grading the malignancy of prostate cancers” because this application was integral part of the publication of Böcking et al., 2015 (attached), in which we have coined the term “Diagnostic/Prognostic DNA-Karyometry”. This new method implies, besides screening for cancer cells, grading of malignancy that therefore should also be dealt with in this review.

Reviewer 2 Report

Here the authors present the development of both the hardware and software of microscopical device for automated detection of cancer cells in oral smears etc. The development of the device may contain important implications for the readership of the journal.

However, there are several minor points to be addressed before being considered for publication in Cancers.

1.     In conclusion, the authors state that with oral smears from Fanconi-anemia patients the device reached an overall diagnostic accuracy of 91.3% when lowering the threshold of abnormal nuclei to >4%, which is shown in Table 2b. However, in simple summary and in abstract, they state the value to be 95.7%. From where this 95.7% came from? Please explain.

2.     There seems to be no Figure 4 in the manuscript.

3.     Please check for typos. (ex. Lane 618: 84.8%% should be 84.8%)

Author Response

Reviewer 2:

  • The mentioned error in mentioning the figure for ODA in “Summary” and “Simple summary” has been corrected. I apologize.
  • Figure 4 indeed was included in our submission and in the galley proof that I have received.
  • The mentioned typing error has been corrected.

Reviewer 3 Report

Authors should further clarify and elaborate novelty in their abstract.

The introduction is deprived of the related work with the recent literature.

There are several interesting papers that look into machine and Deep learning in healthcare. For instance, the below papers has some interesting implications that you could discuss in your Introduction and how it relates to your work.

Vulli, A.; et al.. Fine-Tuned DenseNet-169 for Breast Cancer Metastasis Prediction Using FastAI and 1-Cycle Policy. Sensors 2022, 22, 2988.

Ali, Farman, et al. "A fuzzy ontology and SVM–based Web content classification system." IEEE Access 5 (2017): 25781-25797.

In the introduction, what is the recent knowledge gap of the main literature that the author needs to write this research? What we have known and what we have not known? What is missing from current works? Please explain and give examples!

The presentation of the paper is low.

How authors handle overfitting in the data?

What is your proposed method and why?

What are the limitations of the preset study? 

Add more discussion when interpreting the tables.

Future scope is missing; the readers need to follow your suggestions as well.

Author Response

Reviewer 3:

  • Our introduction instead describes all so far commercially available, comparable

instruments and provides respective references.

  • It was not the purpose of this invited review to report on other types of application of machine- and deep learning as cytopathology, like histopathology or pornography!
  • A respective sentence, describing the aim of this invited review has now been introduced at the end of the Introduction.
  • The criticism that “The presentation of this paper is low” is not understood by A.B..
  • A respective sentence, asking for larger studies has now been added at the end of “Discussion”.
  • It had already been described that missing qualified personnel, insufficient diagnostic accuracy of subjective screening for cancer cells resp. low reproducibility of grading the malignancy of prostate cancers and time consuming subjective evaluation ask for AI-based microscopical diagnostic assistance.
  • The limitations of this study concerning the low number of investigated specimens has now been addressed at the end of “Discussion”.
  • Tables 2a and 2b were already described in 35 lines of “Results” and discussed in 18 lines of “Discussion”.
  • Future scope is now addressed at the end of “Discussion”.

Thank you for reviewing our manuscript and best regards.

Round 2

Reviewer 3 Report

.